# Composite subsystem symmetries and decoration of sub-dimensional excitations

Avi Vadali[1], Zongyuan Wang[2], Arpit Dua[2], Wilbur Shirley[3] and Xie Chen[2]

**1** California Institute of Technology, Pasadena, California 91125, USA
**2** Department of Physics and Institute for Quantum Information and Matter,
California Institute of Technology, Pasadena, California 91125, USA
**3** School of Natural Sciences, Institute for Advanced Study,
Princeton, NJ 08540, USA

## Abstract

Flux binding is a mechanism that is well-understood for global symmetries. Given two systems, each with a global symmetry, gauging the composite symmetry instead of individual symmetries corresponds to the condensation of the composite of gauge charges belonging to individually gauged theories and the binding of the gauge fluxes. The composite charge that is condensed is created by a "short" string given by the new minimal coupling corresponding to the composite symmetry. This paper studies what happens when combined subsystem symmetries are gauged, especially when the component charges and fluxes have different sub-dimensional mobilities. We investigate 3+1D systems with planar symmetries where, for example, the planar symmetry of a planon charge is combined with one of the planar symmetries of a fracton charge. We propose the principle of *Remote Detectability* to determine how the fluxes bind and potentially change their mobility. This understanding can then be used to design fracton models with sub-dimensional excitations that are decorated with excitations having nontrivial statistics or non-Abelian fusion rules.

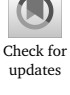

# 1  Introduction

In the past couple of decades, the study of topological phases of matter has moved to the forefront of theoretical condensed matter physics. This has been partly fueled by their applications in quantum error correction. The classification of topological phases of matter in two spatial dimensions in terms of anyon theories, modular tensor categories, and chiral central charges forms the cornerstone of the subject. In three spatial dimensions, there exist phases beyond the conventional description based on topological quantum field theories (TQFT). These are fracton topological phases, and they present a challenge to the classification of phases in three dimensions and higher. Unlike topological order, obtained by gauging global (0-form) symmetries, fracton models are obtained by gauging rigid subsystem symmetries and are characterized by emergent quasiparticles with restricted mobilities. We know several examples of fracton models, many similar in their properties to the canonical examples of the X-cube model and the Haah code [1,2]. Fracton models with more exotic properties, such as non-Abelian superselection sectors, are also known. However, the space of fracton models is not entirely understood. Hence, we need more ways to build and understand fracton models. In this work, we show how exotic fracton models can be constructed by gauging composite symmetries that are subgroups of global symmetries of models, possibly supported in different spatial dimensions. For instance, we consider a model that is the combination of a 3D plaquette Ising Model and a stack of 2D Ising models, and we can gauge the combined global subsystem symmetry of a plaquette Ising model with global symmetries of layers.

In gauge theories of global symmetries, it is well understood that if we start with two independent global symmetries and condense the composite of the two gauge charges, their corresponding fluxes bind together to remain deconfined in the condensate. Consider, for example, two 2D planes, each with a global $\mathbb{Z}_2$ symmetry, as shown in Fig. 1. When the planes are coupled to $\mathbb{Z}_2$ gauge fields separately, they each have a $\mathbb{Z}_2$ gauge charge ($e_1$ and $e_2$) and a corresponding $\mathbb{Z}_2$ flux ($m_1$ and $m_2$). If, instead, only the composite global symmetry is gauged (the two planes are coupled to the same gauge field), the composite of the two symmetry charges $e_1 e_2$ is no longer a symmetry charge. In the gauge theory, this corresponds to the condensation of the gauge charge pair. As a result, individual gauge fluxes become confined, while the flux pair remains deconfined, meaning that $m_1$, and $m_2$ always appear together. Such a mechanism is, of course, well understood and plays a vital role in, for example, coupled layer construction of 3D models such as the 3D toric code and twisted gauge theories.

What happens if we have subsystem symmetries instead of global 0-form symmetries? For simplicity, we will focus on planar symmetries in 3D systems in this paper. There are several possibilities. First, the symmetry charges can be planons, lineons, fractons, which are point excitations that move in 2D, 1D, and 0D submanifolds in the system. The planons, lineons,

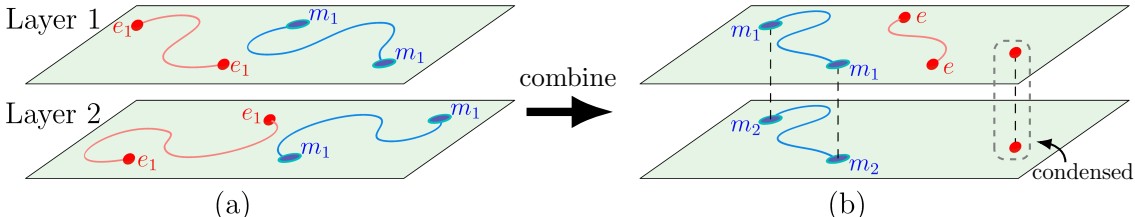

Figure 1: Gauging two planes with $\mathbb{Z}_2$ global symmetries. (a): gauging the two layers separately. We have two copies of the $\mathbb{Z}_2$ gauge theory with planon charges $e_i$ and planon fluxes $m_i$. (b): gauging the combined symmetry of the two planes. The gauge fluxes $m_1$ and $m_2$ now bind together, becoming a new planon flux. Individual charge $e_1$ or $e_2$ remains deconfined planon excitations while the charge pair $e_1 e_2$ is condensed.

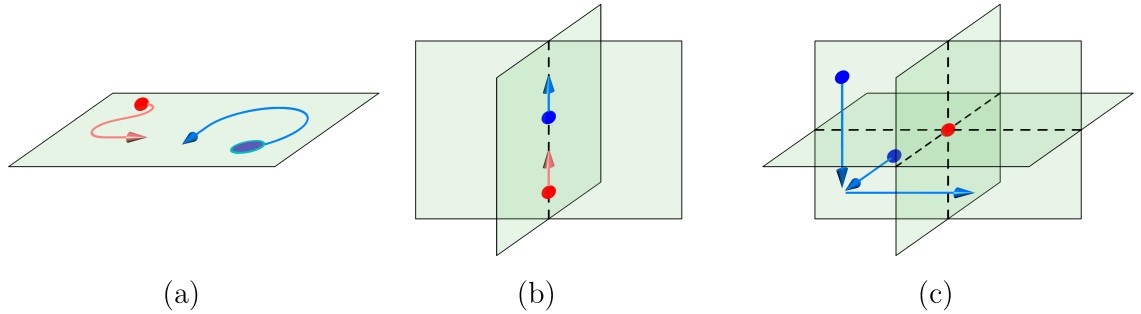

Figure 2: Gauging correspondence for planar subsystem symmetries. (a) planon charge (red, transforming under one planar symmetry) and the corresponding planon flux (blue); (b) lineon charge (transforming under two planar symmetries) and the corresponding lineon flux; (c) fracton charge (transforming under three planar symmetries) and the corresponding lineon fluxes.

and fractons transform under one, two, and three sets of (intersecting) planar symmetries, respectively, as shown in Fig. 2. When the symmetries are gauged, the corresponding gauge flux of a planon charge is a planon in the same plane; the gauge flux of a lineon charge is a lineon along the same line; the gauge fluxes of a fracton charge are lineons in $x$, $y$, $z$ directions that fuse into identity, as shown in Fig. 2. This set of correspondence was shown explicitly in Ref. [3]. When we combine the subsystem symmetries of different charges, there are several potential outcomes. One possibility is when we have charges of the same type, and all their overlapping planar symmetry generators are respectively combined to give the new symmetries. For example, given two fracton charges, we can combine their planar symmetries in $xy$, $yz$, $zx$ planes, respectively. If separately gauged, the two charges correspond to the same type of flux, and it is natural to expect that when such a combined symmetry is gauged, the two fluxes bind together to give the new flux with the same mobility. The second possibility is a composite symmetry combination, which happens when we have charges of potentially different types such that not all of their planar symmetry generators can or are one-to-one combined. For example, we may have a fracton charge with planar symmetry in $xy$, $yz$, and $zx$ planes and a planon charge in the $xy$ plane, and their $xy$ planar symmetries are combined. If gauged separately, the fracton charge corresponds to lineon fluxes in $x$, $y$, and $z$ directions, while the planar charge corresponds to planar flux in the $xy$ plane. When the combined symmetry is gauged, what is the new flux like? Do the original $x$, $y$, $z$ lineon, and $xy$ planon fluxes bind into new lineon fluxes, planon fluxes, or fracton fluxes?

We address this problem in this paper. We find that the way the original fluxes bind to give the new fluxes depends not only on the mobility of the original fluxes and the plane where symmetries are combined but also sensitively on how the symmetries are combined exactly. In particular, we are going to show in section 2, two examples, both of which start with one set of fracton charges and three sets of planon charges in $xy, yz, zx$ planes, and the three sets of planar symmetries of the fracton charge are respectively combined with the planar symmetries of the planon charges. However, the way the original lineon fluxes and planon fluxes bind together to give the new fluxes are very different. In Model FP1, two original planon fluxes from intersecting planes attach to an original lineon flux along the intersection line to give a new lineon flux. In Model FP2, the lineon flux disappears, and the lineon dipole binds with an original planon flux to give a new planon flux. We obtain this result by solving the gauge theory lattice model. Can we arrive at the result without doing lattice level calculation? In section 3, we use the principle of "remote detectability": *For fractional (charge) excitations, there exist operators that detect their existence at a large distance; for non-fractional (charge) excitations, no operator can detect their existence at a large distance.* Applying this principle to the subsystem symmetry cases we are interested in, we can see directly why Models FP1 and FP2 work differently, as described above. This understanding is helpful because we can design models where subsystem symmetry fluxes are bound to planon fluxes in specific ways and acquire nontrivial statistics or non-Abelian features through the process. We discuss in section 4, how lineon and fracton fluxes can be decorated in this way, reproducing nontrivial fracton order like semionic X-cube, Ising Cage-net, etc. Section 5 summarizes the paper.

## 2  Fracton + planon: Two examples model FP1 and model FP2

In this section, we present examples of 'gauging composite symmetry' and demonstrate the phenomenon of flux binding. We present two classes of models, Model FP1 and Model FP2. The construction of these models uses a fracton-charge system and stacks of planon-charge systems. The former is associated with $\mathbb{Z}_2$ fracton charges transforming under $xy, yz$, and $zx$ planar symmetries, and the latter is associated with $\mathbb{Z}_2$ planon charges that transform under the planar symmetries of the layers in the stacks. We consider the stacks of planon-charge systems along three directions, i.e., the $xy, yz$, and $zx$ planes. Thus, in both the fracton-charge and planon-charge systems, we have a planar symmetry associated with each lattice plane. We now consider a symmetry of the combined system generated by products of planar symmetries of the fracton-charge system and the planon-charge system. In particular, in each $xy$ plane, the planar $\mathbb{Z}_2$ symmetry associated with the fracton charge is combined with the $xy$ planar $\mathbb{Z}_2$ symmetry associated with the $xy$ planons to yield a symmetry generator of the combined system. The symmetry generators in the $yz$ and $zx$ planes are defined similarly for the combined system. We refer to this symmetry of the combined system as the composite symmetry. This general description of the composite symmetry holds for both Model FP1 and Model FP2. However, as we will see below, the exact structures of the planar-charge system and the planar symmetry are different for Model FP1 and Model FP2. This leads to a difference in the gauge fluxes of Model FP1 and Model FP2.

For both examples, we start from the paramagnetic product state where all matter qubits are in the $|+\rangle = \frac{1}{\sqrt{2}}(|0\rangle + |1\rangle)$ state. Our goal is to study the exact solvable model obtained by gauging the composite symmetry [3,4] and compare it to the models obtained by gauging the individual symmetries of the fracton-charge and planon-charge systems separately; in particular, we observe how the fluxes of the model obtained by gauging the composite symmetry are a composite of the fluxes of the decoupled gauged models. For instance, gauging the fracton-charge system alone gives a model with lineon fluxes, while gauging the planon-

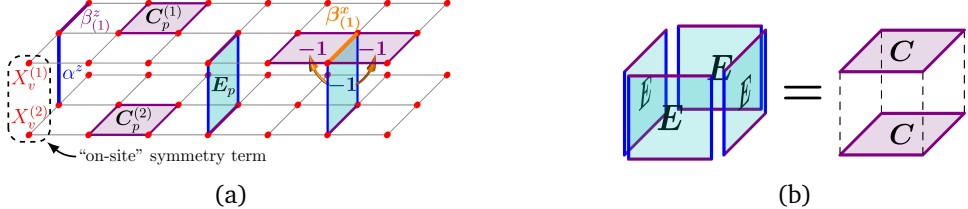

Figure 3: Bilayer Planon model. (a): from left to right, the "on-site" symmetry term $X_v^{(1)}X_v^{(2)}$; the minimal coupling terms and the flux terms $\mathbf{C}^{(1)}$, $\mathbf{C}^{(2)}$ and $\mathbf{E}$. The $\mathbf{E}$-type flux term does not contribute to superselection sectors because its excitation can be converted into $\mathbf{C}$-type excitations on either the top or the bottom plane by the action of a Pauli-$X$ operator, e.g. $\beta_{(1)}^x$. Hence, we can set $\mathbf{E}_p = \mathbb{I}$. (b): Consider the product of four $\mathbf{E}$ terms. The same product is equal to that of $\mathbf{C}_p^{(1)}$ and $\mathbf{C}_p^{(2)}$. Hence, these two $\mathbf{C}$ terms must be simultaneously excited since $\mathbf{E}_p = \mathbb{I}$. Notice that the $\mathbf{C}$ terms are exactly the original flux terms had we gauged the planes individually. Constructing the string operator, we find that the original fluxes must bind together to remain deconfined.

charge system alone gives planon fluxes. In Model FP1 (Sec. 2.2) obtained by gauging the composite symmetry, the lineon flux of the gauged fracton charge system binds with two planon fluxes of the gauged planon charge system to form a new lineon flux. In Model FP2 (Sec. 2.3) obtained by gauging the composite symmetry of a different nature, each lineon flux gets confined, but a lineon dipole binds with a planon flux to form a new planon flux.

Below, we first illustrate the procedure of gauging the composite symmetry with a simple example. In particular, we consider a 2D bilayer Ising paramagnet in section 2.1, such that two global planar symmetries of the two layers can be combined into a planar composite symmetry (see Fig. 1) and gauged. Following that, we discuss Model FP1 and Model FP2 in section 2.2 and section 2.3 respectively.

## 2.1 Warm-up: Bilayer planon model

We consider a 2D bilayer Ising system, consisting of two layers of 2D Ising paramagnets, each with a $\mathbb{Z}_2$ on-site global symmetry $\prod_v X_v^{(1|2)}$ where 1 and 2 are the layer indices. On gauging the planar symmetry of each layer individually, the minimal couplings are intralayer 2-qubit terms, which we denote as $\beta$. The resulting pure gauge theory is just two decoupled copies of the 2D Toric Code, with the Hamiltonian,

$$H = -\sum_{\ell \in \{1,2\}} \sum_{p \in P_\ell} \mathbf{C}_p^{(\ell)} - (\text{charge terms}), \tag{1}$$

where $\ell$ is a layer index, $P_\ell$ denotes the plaquettes in each layer and $\mathbf{C}$ denote the 4-qubit flux terms of $\beta$ gauge qubits around a plaquette; see Fig. 3(a). Here, we write only the flux terms in notation form, as our goal is to demonstrate flux binding.

We are now interested in gauging the composite symmetry $\prod_v X_v^{(1)}X_v^{(2)}$ of the bilayer system. For this composite symmetry, we again have the 2-qubit intralayer minimal couplings in each plane; we denote these as $\beta$. However, there is now an additional 2-body coupling that acts on one qubit on the bottom layer and one qubit on the top layer; we label these as $\alpha$. For gauging, we add the corresponding $\beta$ and $\alpha$ gauge qubits on the edges defining the minimal couplings. We get the aforementioned $\mathbf{C}$-flux terms from the $\beta$ minimal couplings. Due to the additional minimal coupling, a new flux term arises, a 4-qubit term acting on two $\alpha$ gauge qubits and two $\beta$ gauge qubits; we dub these the $\mathbf{E}$-flux terms.

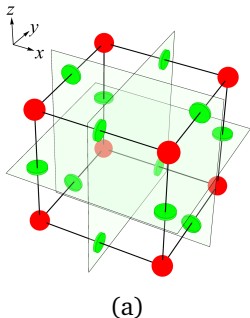 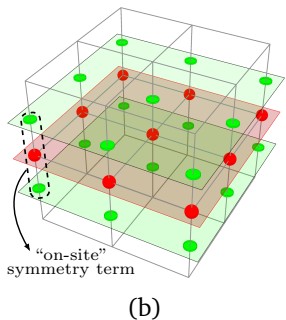

(a) (b)

Figure 4: Model FP1. (a): lattice setup. We consider a 3D Ising paramagnet with planar subsystem symmetries on a cubic lattice with the qubits living on the vertices (red spheres). For the stacks of the 2D Ising paramagnet systems (indicated by the planes) with 2D global symmetry, we place the qubits (green disks) at the centers of the edges. (b): the sandwiched structure of the composite symmetry in Eq. (3).

We integrate out the original matter qubits, i.e., by setting their states to $|+\rangle$ in this case, to obtain a pure gauge theory. This pure gauge theory has one vertex term per matter spin and two kinds of flux terms **C** and **E**, see (a) of Fig. 3. The Hamiltonian is given by

$$H = -\sum_{\ell \in \{1,2\}} \sum_{p \in P_\ell} \mathbf{C}_p^{(\ell)} - \sum_{p'} \mathbf{E}_{p'} - (\text{charge terms}), \tag{2}$$

where $P_\ell$ stands for the planes and $p'$ denotes plaquettes connecting the two layers. As mentioned, the **C**-type flux terms are also the flux terms obtained from gauging the individual symmetries of the layers. In contrast, the **E** flux terms are purely due to the gauging of the composite symmetry.[1]

We now observe that the flux in the model given by Eq. 1 is a composite of the original fluxes. We see this through a relation between the original fluxes and new fluxes. We consider an excitation of a single $\mathbf{E}_p$ flux; by applying a Pauli X operator, we can convert this single $\mathbf{E}_p$ excitation to either a pair of $\mathbf{C}_p^{(1)}$ or a pair of $\mathbf{C}_p^{(2)}$ excitations (see Fig. 3(a)). Hence, the **E** terms do not contribute to the superselection sectors. In other words, this equivalence allows us to set the **E**-type fluxes equal to the identity ($\mathbf{E}_p = \mathbb{I}$), so the new **E**-type fluxes give rise to a relation between the **C**-type fluxes: the product of a pair of **C**-type fluxes with one flux in each plane is equivalent to a product of four **E**-type fluxes around the side faces of a cube; see Fig. 3(b). If $\mathbf{E}_p = \mathbb{I}$, the two **C**-type flux terms must be simultaneously excited, as the product of the two **C** fluxes must have eigenvalue +1. In this sense, the two **C** planon fluxes bind together to give the planon flux of the model with gauged composite symmetry.[2]

## 2.2 Fracton + planon: Model FP1

We now introduce our first nontrivial example. As mentioned above, the model is built by combining a fracton-charge system and planon-charge systems and gauging a symmetry of the integrated system generated by composite planar symmetries. Below we show that the gauge theory obtained from gauging the composite symmetry has a lineon flux composed of planon and lineon fluxes from the decoupled gauge theories.

We consider a cubic lattice where the matter qubits of a classical Ising model live on the vertices as shown in (a) of Fig. 4. We call this the fracton-charge system due to the symmetries

---

[1] In our examples, we'll use a choice of gauge fields that also produces the original flux terms; this is a particular choice that is of interest to us because we want to see the relation between the new and original fluxes from the decoupled theories. One could gauge the composite symmetry differently so the original fluxes no longer appear.

[2] This binding can be verified by explicitly writing the string operators for the fluxes.

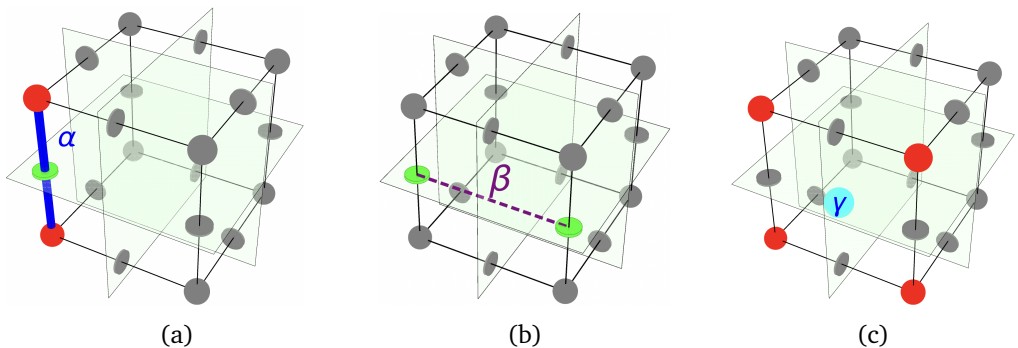

(a)         (b)         (c)

Figure 5: Model FP1: the minimal coupling terms. (a): A 3-qubit minimal coupling term of $\mathcal{S}_{i_\mu}^\mu$. We add a $\mathbb{Z}_2$ gauge qubit to the edge connecting the three matter qubits; we label this gauge qubit as $\alpha$ corresponding to the $\alpha$-coupling. The charges associated with the $\alpha$-coupling are the 'composite condensate' of this model. It requires a planon charge to fuse with a fracton dipole into the topological vacuum. (b): A 2-qubit minimal coupling term of $\mathcal{S}_{i_\mu}^\mu$. The $\beta$ gauge qubit lives on the edge shown by the dashed line. Apart from a directional label, $\beta$ has a plane label in the square brackets. We suppress the position index of the plane. The $\beta$'s are also the original minimal coupling terms for the planon-charge systems. (c): A 4-qubit minimal coupling term of the fracton-charge system. $\gamma$, however, is not a minimal coupling term for $\mathcal{S}_{i_\mu}^\mu$.

we consider for this model. The fracton-charge system has planar subsystem symmetries given by $\prod_{\nu \in P_{i_\mu}^\mu} X_\nu$ for every plane $P_{i_\mu}^\mu$ with $\mu \in \{x, y, z\}$ and $i_\mu$ being the position index of the plane. We then insert three stacks of Ising paramagnets (planon-charge systems) such that their qubits are located on the edges. Each planon-charge system has a 2D global symmetry given by $\prod_{\rho \in P_{(i_\mu, i_\mu+1)}^\mu} X_\rho$ where $(i_\mu, i_\mu+1)$ indicates the edges on which the qubits of the planon-charge systems live. We gauge the composite subsystem symmetry whose generators as shown in Fig. 4 (b) are given by

$$\mathcal{S}_{i_\mu}^\mu = \prod_{\rho \in P_{(i_\mu, i_\mu+1)}^\mu} \prod_{\nu \in P_{i_\mu}^\mu} \prod_{\rho' \in P_{(i_\mu-1, i_\mu)}^\mu} X_\rho X_\nu X_{\rho'} . \tag{3}$$

Recall each symmetry generator is a "sandwich" involving 2 planon-charge symmetry generators and 1 fracton-charge symmetry generator. Hence taking the product of all symmetry generators orthogonal to a given axis should leave only the product of all fracton symmetry generators orthogonal to that axis. Hence we notice a relation among the symmetry generators of this model.

$$\prod_{i_x} \mathcal{S}_{i_y}^x = \prod_{i_y} \mathcal{S}_{i_y}^x = \prod_{i_z} \mathcal{S}_{i_z}^x . \tag{4}$$

We note that this composite symmetry group for Model FP1 is isomorphic to the symmetry group of the undecorated plaquette Ising model.

We consider three types of coupling terms associated with $\mathcal{S}_{i_\mu}^\mu$. There are the 3-qubit coupling terms (red and green qubits shown in Fig. 5(a)), which we label as $\alpha$. There are the 2-qubit coupling terms (green qubits shown in Fig. 5(b)), which we label as $\beta$. Lastly, there are the 4-qubit coupling terms (red qubits shown in Fig. 5(c)), which we label as $\gamma$. Even though the $\gamma$ couplings are generated by the $\alpha$ and $\beta$ coupling terms, we consider them in our gauging process since we are interested in demonstrating how the fluxes from decoupled

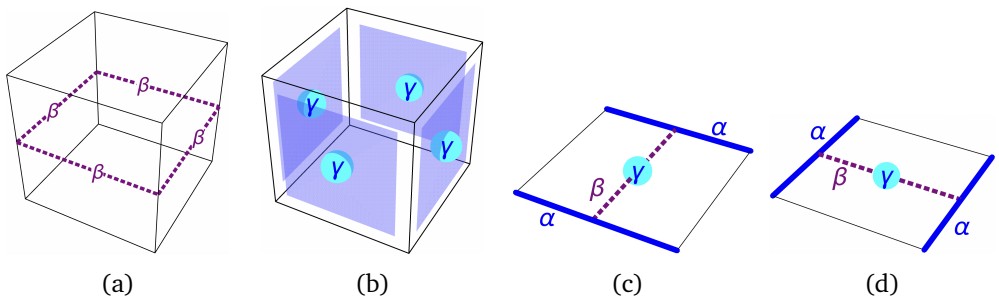

|  |  |  |  |
|:-:|:-:|:-:|:-:|
| (a) | (b) | (c) | (d) |

Figure 6: Model FP1: The three types of flux terms: $\mathbf{C}_p^\mu$, $\boldsymbol{\Gamma}_p^\mu$, and $\mathbf{E}_p^{[\mu](\nu)}$. (a) & (b): A $\mathbf{C}_p^z$ flux term and a $\boldsymbol{\Gamma}_p^z$ flux term. They are those of the 2D planon-charge and the 3D fracton-charge systems respectively. (c) & (d): there are two kinds of $\mathbf{E}$-type flux terms on the face of a cube of the cubic lattice. Drawn here are (c) $\mathbf{E}_p^{[z](y)}$ and (d) $\mathbf{E}_p^{[z](x)}$ on the same face $p$.

gauged models bind together and form fluxes for the model obtained by gauging the composite symmetry.

To gauge the composite symmetry, we add a gauge qubit for each type of coupling term [3, 4] as shown in Fig. 5. We label the gauge qubits as $\alpha$, $\beta$, and $\gamma$ corresponding to the minimal couplings they are associated with. Following the gauging procedure, we obtain the Hamiltonian for Model FP1 as,[3]

$$H^{FP1} = -\sum_{\mu,\{i_\mu\}}\left(\sum_{p\in P_{i_\mu}^\mu}\boldsymbol{\Gamma}_p^\mu + \sum_{p\in P_{(i_\mu,i_\mu+1)}^\mu}\mathbf{C}_p^\mu + \sum_{\nu,p\in P_{i_\mu}^\mu}\mathbf{E}_p^{[\mu](\nu)}\right) - (\text{charge terms}). \tag{6}$$

Here, the $\mathbf{C}$-type flux terms are the original fluxes of the planon-charge systems, and the $\boldsymbol{\Gamma}$-type flux terms are the original flux terms of the fracton-charge system. The $\mathbf{E}$-type flux terms are the new flux terms that arise in the model obtained from gauging composite symmetries. Similar to the bilayer model, we observe that the $\mathbf{E}$ terms do not contribute to the superselection sectors. For an excitation of the $\mathbf{E}$-type flux term i.e., $\mathbf{E}_p^{[\mu](\nu)} = -1$, we can act a Pauli $X$ operator to turn it into two $\mathbf{C}$-type fluxes with the two corresponding $\mathbf{C}$ terms having the eigenvalue of $-1$. Hence, we can set all $\mathbf{E}_p^{[\mu](\nu)} = \mathbb{I}$. This gives a relation between the $\boldsymbol{\Gamma}$ and the $\mathbf{C}$ terms. The product of four $\mathbf{E}$ terms around the side faces of a cube must be identity, but the product is also equal to the product of a $\boldsymbol{\Gamma}$ and a $\mathbf{C}$ flux terms; see Fig. 7. The lineon string operator that satisfies this set of relations and the trivial $\mathbf{E}$-flux condition is described in Fig. 7. We observe that the string operator is the product of a lineon flux string operator from the fracton-charge system, together with two planon flux string operators from two intersecting planon-charge systems. Thus, we conclude that the new lineon flux is the composite of a lineon flux and two planon fluxes from the decoupled models, respectively.

## 2.3 Fracton + planon: Model FP2

We present another nontrivial example of a model obtained from gauging composite symmetry. The underlying models consist of the same fracton-charge systems and the planon-charge

---

[3]As mentioned earlier in footnote 1, this is a particular choice of gauging of interest to us. We can make a different choice of gauging such that the fluxes of the gauged Hamiltonian $H^A$ come from only the $\alpha$ and $\beta$ couplings as

$$H^A = -\sum_{\mu,\{i_\mu\}}\left(\sum_{p\in P_{(i_\mu,i_\mu+1)}^\mu}\mathbf{C}_p^\mu + \sum_{p\in P_{i_\mu}^\mu}\mathbf{B}_p^\mu\right) - (\alpha,\beta \text{ charge terms}), \tag{5}$$

where $\mathbf{B}_p^\mu = \mathbf{E}_p^{[\mu](\nu)} \times \mathbf{E}_p^{[\mu](\rho)}$ is the product of two $\mathbf{E}$ terms on the same plaquette $p$.

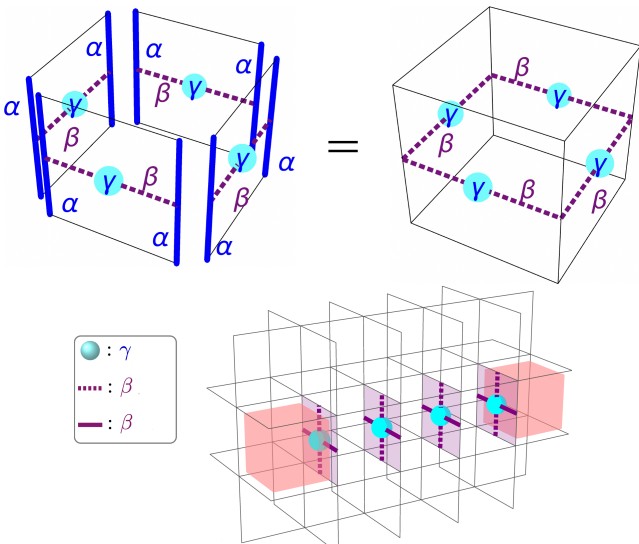

Figure 7: Model FP1: one of the relations between the original flux terms. The remaining relations can be obtained by rotations. LHS: the relation is the product of the four $\mathbf{E}$-type flux terms as shown, two $\mathbf{E}_p^{[y](x)}$'s and two $\mathbf{E}_p^{[x](y)}$'s. RHS: the $\mathbf{C}$-type and $\boldsymbol{\Gamma}$-type flux terms also have a relation. Right: Model FP1: the new lineon string operator for a $y$-mobile lineon. The locations of the new lineon excitations are indicated by the pink cubes. From this string operator, we see clearly that the new lineon is a composite of the original lineon flux together with two original planon fluxes from perpendicular planes.

systems as considered for Model FP1. However, the planon-charge systems are laid out such that the matter qubits live on the vertices; thus the nature of the composite symmetry is different. Below we show that the gauge theory obtained from gauging the composite symmetry has only planon fluxes; each is a composite of a planon flux and a lineon dipole flux from the decoupled gauged theories.

Similar to Model FP1, we have the fracton-charge system with matter qubits on the vertices. We insert the three stacks of planon-charge systems into the cubic lattice, as shown in Fig. 8(a). The plaquettes of the planon-charge systems (not drawn) coincide with those of the cubic lattice. Each vertex hosts four matter qubits, one (the red sphere) from the fracton-charge system and three (the green disks) from three mutually intersecting planon-charge systems. We gauge the composite symmetry whose generators are given by

$$\mathcal{S}_{i_\mu}^\mu = \prod_{v \in P_{i_\mu}^\mu} X_{v,f} X_{v,\mu}\,, \tag{7}$$

where $f$ denotes the matter qubit belonging to the fracton-charge system and $\mu$ denotes the matter qubit belonging to the planon charge system in the plane with direction $i_\mu$. We note that the composite symmetry group of Model FP2 is isomorphic to that of 3 decoupled stacks of planon symmetry generators with no global relation.

The couplings associated with the composite symmetry are the 2-qubit $\beta$-coupling terms, the 4-qubit $\gamma$-coupling terms, and the 2-qubit $\alpha$-coupling terms as shown in Fig. 8. We add gauge qubits for each coupling term. Following the gauging procedure, we obtain the Hamiltonian for Model FP2 as

$$H^{FP2} = -\sum_{\mu,\{i_\mu\}} \left( \sum_{p \in P_{i_\mu}^\mu} \boldsymbol{\Gamma}_p^\mu + \sum_{p \in P_{i_\mu}^\mu} \mathbf{C}_p^\mu + \sum_{v,p \in P_{i_\mu}^\mu} \mathbf{E}_p^{[\mu](v)} \right) - (\text{charge terms})\,. \tag{8}$$

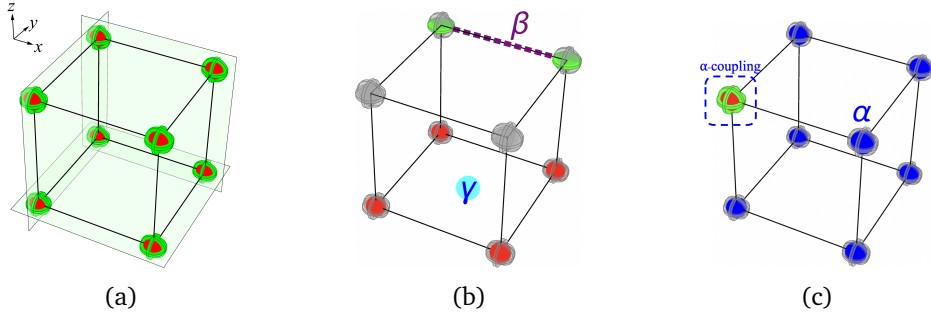

Figure 8: Model FP2. (a): lattice setup. Same as in Model FP1, we place the matter qubits of the fracton-charge system on the vertices. The stacks of planon-charge systems are arranged such that their qubits also live on the vertices. The plaquettes of the planon-charge systems coincide with those of the cubic lattice. (b): the original minimal coupling terms. The 2-qubit $\beta$-coupling terms are also minimal coupling for the composite symmetry Eq. (7). (c): the new minimal coupling term as a result of gauging the composite symmetry.

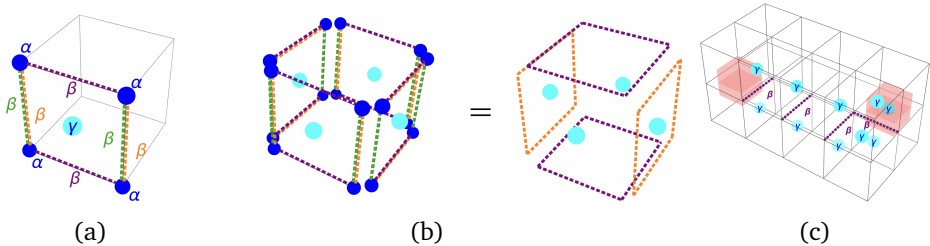

Figure 9: Model FP2. (a): One of the **E**-type flux terms ($\mathbf{E}_p$). The other **E**-type flux terms are plaquettes of the same form in the other 2 spatial planes. (b): One of the relations between the original flux terms. The remaining relations can be obtained by rotations. LHS: the relation is the product of the four **E**-type flux terms as shown, two $\mathbf{E}_p^{[y](x)}$'s and two $\mathbf{E}_p^{[x](y)}$'s. RHS: the **C**-type and **Γ**-type flux terms also have a relation. (c): The new planon string operator for a $z$-plane mobile planon. We indicate the locations of the new planon excitations by the pink cubes. Notice that this string operator is exactly the product of an original $z$-plane planon flux string operator and an original $z$-plane lineon dipole flux string operator.

The fluxes consist of the original flux terms of the decoupled models, i.e., the **C**-type and **Γ**-type flux terms similar to Model FP1. But we also have the new **E**-type flux terms, Fig. 9, that arise from gauging the composite symmetry. Similar to the bilayer Model and Model FP1, we observe that the **E** terms do not contribute to the superselection sectors. In fact, on setting $\mathbf{E}_p^{[\mu]}$ to Identity, we get a relation between the **C** and the **Γ** flux terms. The corresponding gauge flux is a planon composed of a planon flux from the planon-charge system and a lineon dipole flux from the fracton-charge system; see the string operator for this combined gauge flux in Fig. 9.

We note that Model FP2 is equivalent to stacks of 2D toric codes. To see this, we recall that the gauge charges in Model FP2 consist of one fracton and three planons (in the $xy, yz, zx$ directions). We write this charge basis as $\{f, p_{xy}, p_{yz}, p_{zx}\}$. The new minimal coupling (see Fig. 8(c)) corresponding to the composite symmetry is a product of the four elements of the charge basis. Hence $f \times p_{xy} \times p_{yz} \times p_{zx} = 1$, which implies that $p_{xy} \times p_{yz} \times p_{zx} = f$. This suggests that the charge basis is not independent, as the planon charges ($\{p_{xy}, p_{yz}, p_{zx}\}$) generate the fracton charge ($f$). Thus, we can choose an independent charge basis consisting

of only planons ($\{p_{xy}, p_{yz}, p_{zx}\}$). For each basis planon, we can apply an entanglement renormalization circuit to extract a toric code layer [5, 6], leading to the result that Model FP2 is just a stack of 2D toric codes. Thus, we see that different patterns of flux binding, due to gauging different composite symmetries, result in different topological orders.

## 3 Flux binding via remote detectability

In the previous section, we saw through exactly solvable stabilizer models how the lineon fluxes of a fracton charge bind with the planon fluxes of planon charges when some composite planar symmetries of the fracton and planon charges are gauged. In this section, we arrive at the same result without solving the lattice models. In particular, we will use the principle of "remote detectability" to deduce how the old fluxes bind together to form new fluxes. This argument does not rely on the Model FP2 being a stabilizer or even exactly solvable and can be applied generally. We will apply the resulting insight to reproduce some interesting fracton models and construct new ones with nontrivial features in the next section.

The principle of "remote detectability" says: *fractional excitations can be detected with some 'remote' unitary operators which act only at a large distance from the excitation; non-fractional excitations, on the other hand, cannot be detected remotely.*

In this paper, we apply this principle to fractional charge excitations in a gauge theory and deduce the corresponding form of the gauge flux.[4] In particular, knowing the form of the gauge charge, we can deduce the form of the remote detection operator. Truncating the remote detection operator then exposes the shape and mobility of the gauge flux excitation. We illustrate this line of argument using models with 2D global symmetry and 3D planar subsystem symmetries, respectively in section 3.1. Moreover, if we start with two independent systems but only keep some of their composite symmetries, certain local composites of the symmetry charges will no longer carry nontrivial charges. When the composite symmetries are gauged, remote detection operators that detect such local composites no longer give rise to deconfined flux excitations once truncated. We will see in section 3.2, 3.3 how this mechanism leads to the binding of individual fluxes when composite symmetries are gauged.

### 3.1 Flux excitations from truncating remote detection operators

Consider first a 2D system with global $\mathbb{Z}_2$ symmetry. A single charged particle carries a nontrivial symmetry charge while a charge pair does not. After coupling to the gauge field, the remote defection operator, therefore, should be able to detect a single charge but not a pair. This can be achieved, of course, with a loop integration of an electric field that intersects and anti-commutes with the Wilson line connecting to the single charge when it is created, as shown in Fig. 10(c). When the loop-shaped remote detection operator is truncated, it leaves two endpoints (Fig. 10(d)), which correspond to a pair of point flux excitations that can move along the loop operator that twists and turns freely on the 2D plane. The flux excitations are hence planons.

In a 3D model with planar $\mathbb{Z}_2$ symmetry in each $xy$, $yz$, and $zx$ plane and one charge at each triple intersection point of three perpendicular planes, local symmetric processes create a minimum of four charges at a time (Fig. 11 (a)). The remote detection operator should be able to detect not only single charges but also charge pairs remotely. This is achieved with the wireframe operator in Fig. 11 (b). When the wireframe encloses a single charge, it

---

[4]Although remote detection operators do not necessarily act as pure $U(1)$ phases (in the case of non-Abelian anyons with degeneracies), in this paper we only consider systems in which remote detection operators act as $U(1)$ phases on the excited states.

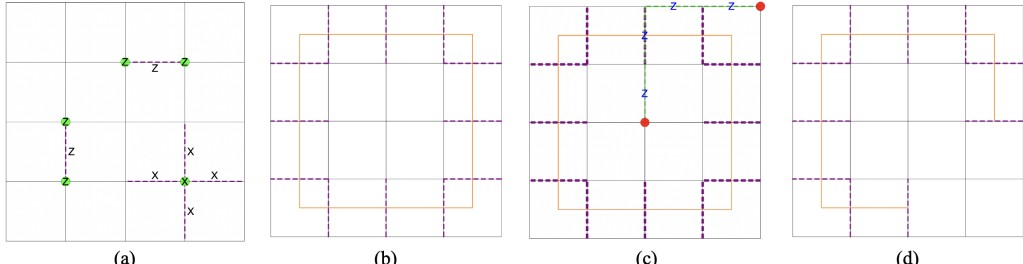

Figure 10: 2D model with global symmetry. (a): Minimal couplings and gauge symmetry operator. Qubits (green disks) live on vertices, and gauge fields (purple edges) are assigned to edges. (b): Operator that detects charges within the bulk (in the ungauged theory). (c): Remote detection operator capable of detecting excitations within the loop (shown in orange). This RD operator is formed by taking products of gauge symmetry operators such that the gauge fields in the bulk cancel out. (d): Loop RD operator detecting anyon charge via anticommutation between the loop and string creation operator. (e): Truncated remote detection operator. The excitations at the boundary of the truncation possess the same mobility as flux excitations.

intersects and anti-commutes with the Wilson 'membrane' that extends from the single charge and detects its existence. The same configuration can be used to detect charge pairs. For example, to detect the vertical pair of charges on the left in Fig. 11 (c), we can use a wireframe operator such that half of the charge pair is 'inside' the wireframe while the other half is outside. Although the two charges in the pair can be very close to each other, this is still a 'remote' detection method because the wireframe operator only acts along the edges of the cube area it encloses and never gets close to either of the charges. Truncating the wireframe operator exposes point flux excitations that move along rigid wireframe edge directions (Fig. 10(d)). The flux excitations are hence lineons, and lineons in the $x$, $y$, and $z$ directions fuse into the vacuum.

To determine the principle of truncation of remote detection (RD) operators, we find and slice the RD operators of a 2D model with global symmetry and a 3D system with planar symmetry and fracton charge (the ingredient models in our construction of sample models with a gauged composite symmetry). We first consider the 2D system, whose remote detection operator is a loop consisting of products of charge operators shown in Fig. 10(a). To determine the exact product of charge operators needed to create the RD loop, we refer to Fig. 10(b). The operator depicted in this figure detects charge excitations within the bulk in the ungauged theory; if there is an odd number of violations (eigenvalue -1) within this membrane of Pauli $X$ operators, the operator is measured to be $-1$ (hence it can detect violations). Thus adding the gauge fields associated with each qubit in Fig. 10(b) should give the gauged RD operator (see Fig. 10(c)). An equivalent method of determining the particular form of the RD operator is to note what product of charge operators annihilates all gauge fields in the bulk and leaves gauge fields only at the region's boundary over which the product of terms is taken. Using this, one can see that a product of charge terms in a closed loop satisfies this exact condition; hence the charge RD operator for a 2D system with global symmetry is a loop operator. This loop RD operator can detect individual anyon charges within the loop by anti-commuting with the excitation creation operator (see Fig. 10(d)). Truncating this loop RD operator (see Fig. 10(e)) reveals a single anyon (violation of $Z$ plaquette stabilizer) at each endpoint; we conclude that the flux excitations of the 2D system are anyons. The same analysis can be done for a general 3D model with planar symmetry and a fracton charge.

Looking at the 3D model, the charge terms are given by cube operators (violations of which

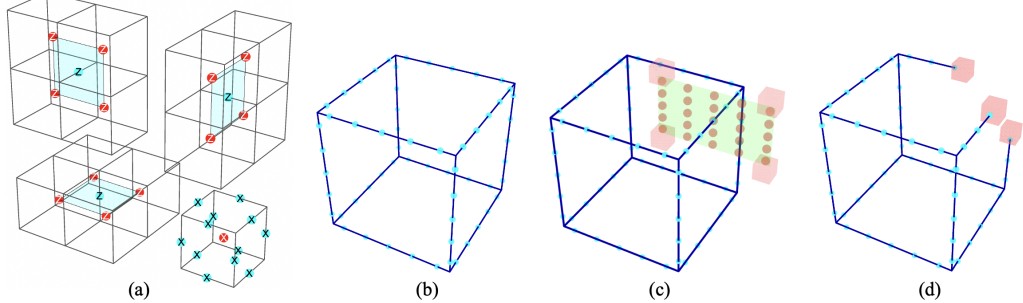

Figure 11: 3D model with planar symmetries and fracton charge. (a): Minimal couplings and gauge symmetry operator. Qubits (red spheres) live at body centers, and gauge fields (cyan spheres) are assigned to edges. (b): Operator that detects charges within the bulk (in the ungauged theory). (c): Remote detection operator capable of detecting excitations within the wireframe (shown in blue). This RD operator is formed by taking products of gauge symmetry operators such that the gauge fields in the bulk cancel out. (d): Wireframe RD operator detecting fracton charge via anticommutation between wireframe and membrane creation operator. This operator detects a fracton dipole in 2 steps: it first detects one fracton and then is moved to detect the second fracton in the dipole. (e): Truncated remote detection operator. The excitations at the boundary of the truncation possess the same mobility as flux excitations.

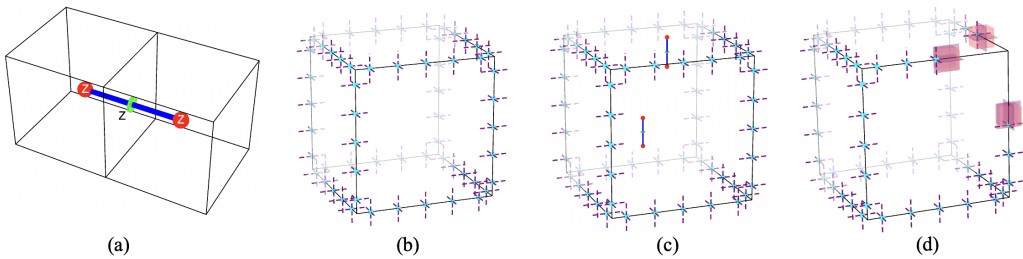

Figure 12: Model FP1. (a): The new 3-body minimal coupling due to gauging the composite symmetry. (b): Remote detection operator capable of detecting excitations within the decorated wireframe (shown in black). This RD operator is formed by taking gauge symmetry operators' products such that the bulk gauge fields cancel out. (c): The RD operator cannot detect the new minimal coupling; hence it is not a fractional excitation. (d): Truncated remote detection operator. The excitations at the boundary of the truncation possess the same mobility as flux excitations.

are fractons) shown in Fig. 11(a). Similar to the 2D model, we can see that in the ungauged theory, to detect a single violation of a Pauli $X$ operator inside a finite volume, we take the product of Pauli $X$'s at the body centers within the cubic volume (see Fig. 11(b)). Adding in the gauge fields according to the cubic charge operator, we see that the gauged RD operator is a cubic wireframe with gauge fields along the edges (see Fig. 11(c)). This wireframe detects fracton charges by anti-commuting with the membrane creation operator (see Fig. 11(d)). Truncating this RD operator (see Fig. 11(e)) shows that the excitations at the endpoints of the wireframe are lineons, i.e., the violations of the $Z$ vertex terms of the 3D system. Additionally, note that since the endpoints of the truncated RD operator are lineons, RD operators can also be viewed as operators that move the flux excitations in a closed path far from the charge.

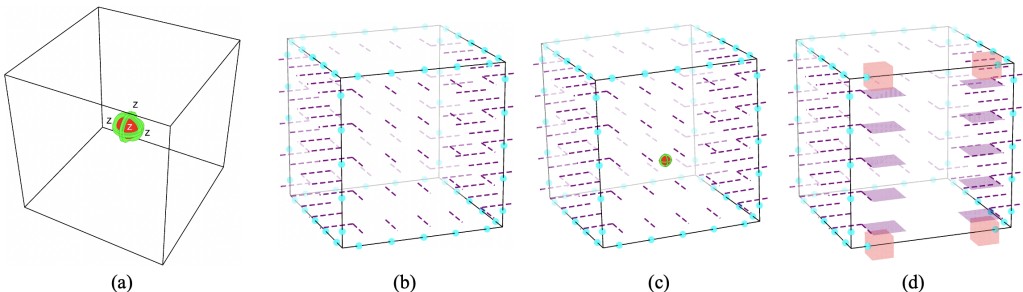

Figure 13: Model FP2. (a): The new 3-body minimal coupling due to gauging the composite symmetry. (b): Remote detection operator capable of detecting excitations within the decorated wireframe (shown in black). This RD operator is formed by taking gauge symmetry operators' products such that the bulk gauge fields cancel out. (c): The RD operator cannot detect the new minimal coupling; hence it is not a fractional excitation. (d): Truncated remote detection operator. The excitations at the boundary of the truncation possess the same mobility as flux excitations.

## 3.2 Remote detection in model FP1

Now we will use the remote detection principle to see how the fluxes bind in Model FP1. Recall that in Model FP1, we had fracton charges at lattice sites of a cube lattice and planon charges on three sets of planes cutting through edges of the cube lattices. If the fracton-charge system is gauged alone, the remote detection operators are in the shape of a wire-frame, and correspondingly, the flux excitations are lineons, as discussed above. If the planon-charge system is gauged alone, the remote detection operators are loops, and correspondingly the flux excitations are planons. If some composite symmetry is gauged, the lineon and planon fluxes must bind together. In particular, according to how the planar symmetries are combined in Model FP1, there is a new type of minimal coupling term containing both the fracton and the planon charges as shown in Fig. 12 (a). The wire-frame and loop remote detection operator would each detect the existence of such a term so we need to combine them in a way that this three-body term is not detected by any remote detection operator as it corresponds to a trivial superselection sector. This can be achieved with a remote detection operator, as shown in Fig. 12(b), where the loop operators are attached to the six surfaces of the wire-frame operator.

The new operator still takes the shape of a wire-frame. We can check that 1. individual fracton charges and fracton dipoles can still be detected if properly placed inside the decorated wire-frame operator; 2. individual planon charges can still be detected if placed on the surface of the decorated wire-frame operator; 3. the new three-body minimal coupling cannot be detected by any of the decorated wire-frame operators, whether the minimal coupling is entirely inside the operator or partially outside as shown in Fig. 12(c). From this operator, we can then determine the form of the new flux excitation. Since the operator still takes the shape of a wire-frame, we can see from its truncation that the flux excitations are still lineons. This conclusion is consistent with that obtained in section 2.2 but applies more generally. In particular, the argument we use in this section is independent of the symmetric state of the fracton and planon charges, so it works even when they form some nontrivial symmetry-protected topological state.

## 3.3 Remote detection in model FP2

A similar argument can be applied to Model FP2. We will see how the difference in the composite symmetry results in a different binding of fluxes compared to Model FP1, such

that the new fluxes are planons instead of lineons. In Model FP2, the fracton charges are still at the lattice sites of a cubic lattice, while the planon charges lie on the $xy$, $yz$, and $zx$ planes of the cubic lattice. The planar symmetries are combined so that there is a new type of minimal coupling term containing the fracton charge and three planon charges at the same lattice site, as shown in Fig. 13 (a). To not detect this term, we need to combine the wire-frame operator with all the loop operators that wind around the wire-frame in a particular direction. For example, Fig. 13 (b) shows one such operator with the wire-frame bound to loop operators in the $xy$ plane. There are two other types of operators with loops in the $yz$ and $zx$ planes, respectively.

We can check that: 1. individual fracton charges and fracton dipoles can still be detected if properly placed inside the decorated wire-frame operator; 2. individual planon charges can still be detected if placed inside the wire-frame operator with loop operators in the same plane; 3. the new four-body minimal coupling cannot be detected by any of the new remote detection operators as shown in Fig. 13 (c). Truncating the remote detection operator exposes the new flux excitations. The operator is not a wire-frame anymore but instead takes the shape of a ribbon loop. When the ribbon is thin, the endpoints of the truncated operator are point excitations that move in planes. Therefore, the elementary flux excitations are planons in $xy$, $yz$, and $zx$ planes, respectively. Again, this result is derived independent of the symmetric state formed by the fractons and planons.

## 4 Decoration of sub-dimensional excitations

The flux-binding result derived in section 3 allows us to construct interesting fracton models by binding lineons / fractons with planons, thereby passing the non-trivial statistics or non-Abelian internal structure of the planons onto the lineons or fractons. We discuss the decoration of lineon excitations and fracton excitations in section 4.1 and section 4.2 respectively. In particular, we demonstrate in the process how some of the Cage-net models discussed in [7] can be obtained by gauging the subsystem symmetry of an SSPT (subsystem symmetry protected topological) model.

### 4.1 Decoration of lineons

To construct the models with the decoration of lineon flux superselection sectors, we follow the construction of Model FP1 (Sec. 2.2) but use the 2D symmetry-protected topological phases (SPT) or symmetry-enriched topological orders (SET) with $\mathbb{Z}_2$ planar symmetry as the planon-charge systems (see Fig. 14). We recall that in Model FP1, the new gauge flux is a composite of the original lineon flux of the fracton-charge system and two planon fluxes from orthogonal planon-charge systems. Below, we consider examples where we replace the 2D Ising paramagnet on the ungauged side by the 2D Levin-Gu SPT with global $\mathbb{Z}_2$ symmetry or the 2D toric code with global $e \leftrightarrow m$ anyon-swap symmetry. We obtain decorated lineon superselection sectors with nontrivial statistics or non-Abelian fusion rules in these cases.

- *Example of model FP1 with decorated lineons that have nontrivial statistics*
  We consider an example of Model FP1 where we introduce nontrivial $\mathbb{Z}_2$ SPTs as the planon-charge systems. In particular, we can use the Levin-Gu SPTs [8] as the planon-charge systems. Gauging just the $\mathbb{Z}_2$ symmetry of the Levin-Gu SPT yields the double semion model that has two semionic fluxes. The flux binding result of Model FP1 implies that upon gauging its composite symmetry, the two semionic fluxes from two orthogonal double semion planes must bind with the lineon flux of the fracton-charge

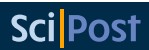

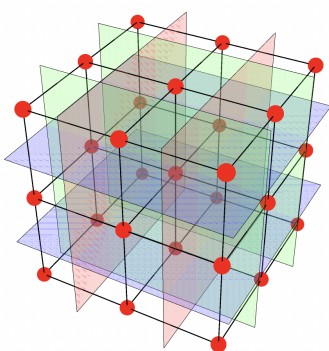

Figure 14: The general setup for Model FP1. We have a fracton-charge system (red spheres) and 2D planon-charge systems (colored planes). The green, blue, and red planes represent the planon-charge system, which are in some potentially non-trivial SPT or SET states.

system to yield a lineon flux composite. The resulting 3D model is the 3D semionic X-cube model [9]. One can show that two lineon fluxes of this model have mutual statistics of $\pi$ by designing processes analogous to braiding in 2D topological orders [9].

- *Example of model FP1 with decorated lineons that have non-Abelian fusion rules*
  We now discuss an example of Model FP1, where we introduce nontrivial SETs as the planon-charge systems. For instance, we consider toric code SETs (enriched by the $e - m$ swap symmetry) and gauge the composite symmetry that is a product of this planar swap symmetry and the planar symmetries of the fracton-charge system. Gauging the swap symmetry alone in the toric code layer would give the doubled Ising string-net model with non-Abelian fluxes, the Ising anyons $\sigma$'s and $\bar{\sigma}$'s, each with quantum dimension $\sqrt{2}$. On gauging the composite symmetry, the non-Abelian Ising anyons from orthogonal doubled-Ising string-net planes must bind with the lineon flux of the gauged fracton-charge system to yield a decorated lineon flux that is non-Abelian with quantum dimension 2. The resulting 3D model is the Ising cage-net model [7].

## 4.2 Decoration of fractons: Model LP

We now consider a different model of flux binding such that we obtain decorated fracton superselection sectors. In particular, we consider a class of models obtained from combining a 3D lineon-charge system with three stacks of 2D planon-charge systems. Hence, we refer to this class of models as Model LP. The key difference between Model LP and Model FP1 is that the construction of the former uses a lineon-charge system, while the latter uses a fracton-charge system instead in its construction. It is due to this change that the gauged model hosts decorated fracton flux superselection sectors instead of decorated lineon flux superselection sectors; the fracton flux is a composite of the original fracton flux with three planon fluxes, each from a different 2D system.

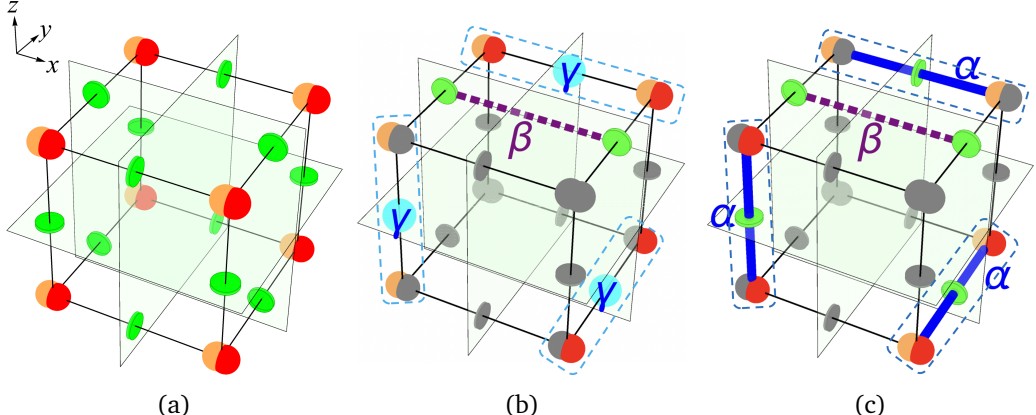

Figure 15: Model LP: lattice setup. We take the 3D lineon-charge system, a model that is dual to the fracton-charge system [3], on the cubic lattice. The orange and red spheres represent the qubits. As in Model FP1 (Sec. 2.2), we insert three stacks of the 2D planon-charge systems bisecting the edges. The green disks show their qubits. (b) & (c): the minimal coupling terms. (a): all three kinds of the original minimal coupling terms $\gamma$ of the 3D lineon-charge system (inside the dashed cyan colored loops), and an example of the minimal coupling of the planon-charge systems $\beta$. (b): the new minimal coupling terms $\alpha$ for the composite symmetry are highlighted by the dashed blue loops. The $\beta$'s and $\gamma$'s remain minimal coupling terms for the composite symmetry.

We demonstrate Model LP's flux binding of fracton flux with planon fluxes by considering a 3D Ising paramagnet with planar symmetries as the lineon-charge system and stacks of 2D models with on-site $\mathbb{Z}_2$ planar symmetries as the planar-charge systems. This lineon-charge system, a 3D Ising paramagnet with planar symmetries, is "dual" to the fracton-charge system discussed earlier; gauging the planar systems in either of these, i.e., the lineon-charge system or the fracton-charge system, yields the X-cube model [3]. The difference in the resulting gauged models is that in the gauging of the lineon-charge system, the X-cube fracton is the flux excitation, while in the gauging of the fracton-charge system, the X-cube lineon is the flux excitation.

We put the lineon-charge system on the cubic lattice with two matter qubits per vertex, the orange and the red spheres in Fig. 15. The orange spins transform under the planar symmetries of the $y$- and $x$-planes, and the red spins transform under the $z$- and $x$-planes. These subsystem symmetries are defined by: $\prod_{v \in P^x_{i_x}} X^{\mathbf{r}}_v X^{\mathbf{o}}_v$ for the $x$-planes; $\prod_{v \in P^y_{i_y}} X^{\mathbf{o}}_v$ for the $y$-planes; and $\prod_{v \in P^z_{i_z}} X^{\mathbf{r}}_v$ for the $z$-planes. We place the qubits of the planon-charge systems on the edges. We recall that each planon-charge system has the 2D global symmetry of $\prod_{\rho \in P^\mu_{(i_\mu, i_\mu+1)}} X_\rho$ where $(i_\mu, i_\mu + 1)$ indicates the edges on which the qubits of the planon-charge systems live. Like Model FP1, the composite symmetry we gauge here also has a sandwiched

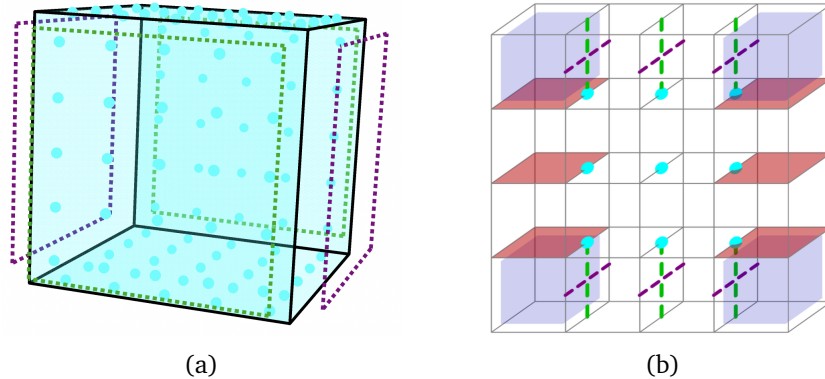

<p style="text-align:center">(a)            (b)</p>

Figure 16: Model LP: flux binding from remote detection operators. (a): A new charge remote detection (RD) operator, which is built from the original RD operators supported on four sides of the cube. The membrane charge RD operator of the lineon-charge system is drawn in cyan color. The green and purple loops are the RD operators of the planon-charge system on the $y$- and $z$-planes, respectively. (b): cutting the charge RD operator reveals a membrane with fractons (purple cubes) at the corners and planons (red plaquettes and green and purple edges) along the sides.

structure. Specifically, for each direction, we consider the symmetry generators,

$$\mathcal{S}^x_{i_x} = \prod_{\rho \in P^x_{(i_x, i_x+1)}} \prod_{\nu \in P^x_{i_x}} \prod_{\rho' \in P^x_{(i_x-1, i_x)}} X_\rho X_\nu^{\mathbf{r}} X_\nu^{\mathbf{o}} X_{\rho'}, \tag{9}$$

$$\mathcal{S}^y_{i_y} = \prod_{\rho \in P^y_{(i_y, i_y+1)}} \prod_{\nu \in P^y_{i_y}} \prod_{\rho' \in P^y_{(i_y-1, i_y)}} X_\rho X_\nu^{\mathbf{o}} X_{\rho'}, \tag{10}$$

$$\mathcal{S}^z_{i_z} = \prod_{\rho \in P^z_{(i_z, i_z+1)}} \prod_{\nu \in P^z_{i_z}} \prod_{\rho' \in P^z_{(i_z-1, i_z)}} X_\rho X_\nu^{\mathbf{r}} X_{\rho'}. \tag{11}$$

We illustrate the minimal couplings associated with the original decoupled symmetries and the new composite symmetry in Fig. 15(b) and Fig. 15(c) respectively. We denote the original minimal couplings (for decoupled symmetries) as $\beta$ and $\gamma$ while the new minimal couplings (for composite symmetries) as $\alpha$. The $\alpha$ terms along different directions $x|y|z$ are specified by adding a subscript $(x)|(y)|(z)$.

    We can determine the flux binding of this model using the principle of Remote Detectability.[5] We recall from Sec. 3 that the closed local membrane operators for the new fluxes in the model obtained by gauging the composite symmetry are the RD operators for the charges. These new RD operators can be constructed using the original RD operators such that they do not detect the 'composite condensates', i.e., the charges associated with the new minimal couplings that were not present for decoupled symmetries. For example, consider the RD operator for charges of the lineon-charge system, drawn in cyan in Fig. 16(a). This operator detects the original $x$ and $y$ lineons. So, it can also detect our 'composite condensates' associated with $\alpha_{(z)}$ and $\alpha_{(y)}$ coupling terms. To prevent the detection of $\alpha_{(z)}$, we add two remote detection operators of the planon-charge system on the side planes (the purple loops). Similarly, for $\alpha_{(y)}$, we add two remote detection operators on the front and back faces, as indicated by the green loops. This yields the required remote detection operator, as stated earlier. By slicing this closed membrane operator, we get the creation operator for the new fracton flux, as shown in Fig. 16(b). The excitation created by this sliced membrane

---

[5]See Appendix A for an alternative explanation of flux binding by writing the gauged model and looking at independent superselection sectors.

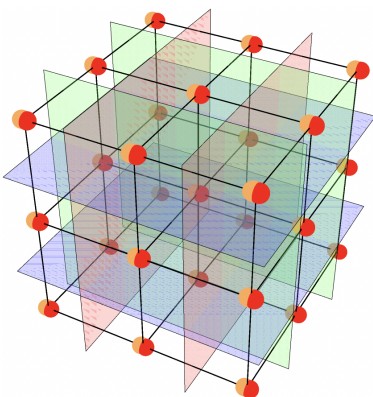

Figure 17: The general setup for Model LP. The Model LP consists of a lineon-charge system and a stack of 2D planon-charge systems. The lineon-charge system has red and orange qubits on vertices, and the 2D planon-charge systems live on dual planes, as shown. The composite symmetry generators act on the qubits of the lineon-charge system and the planon-charge system together.

operator is our bound flux excitation which is a composite of the original fracton fluxes of the gauged lineon-charge system and three planon fluxes from the gauged planon-charge systems in orthogonal layers.

Above, we described a particular example of Model LP class to demonstrate the flux binding that yields decorated fractons. In general, we can consider different planon-charge systems with global $\mathbb{Z}_2$ symmetry instead of the 2D Ising paramagnet. The general setup for Model LP is illustrated in Fig. 17.

We now present an example of ModelLP where we use nontrivial SETs as the planon-charge systems. For instance, we consider 2D toric code SETs (enriched by the e-m swap symmetry) as planon-charge systems. We gauge the composite symmetry, each generator of which is a product of the swap symmetries of two toric code layers and a planar symmetry of the lineon-charge system. As mentioned earlier, gauging the swap symmetry alone in the toric code layer would give the doubled Ising string-net model with non-Abelian planons; the gauge fluxes are the Ising anyons $\sigma$'s and $\bar{\sigma}$'s, each with quantum dimension $\sqrt{2}$. Now, in the case of gauging the composite symmetry, the fractons in the model obtained from gauging composite symmetry are decorated with these non-Abelian anyons and are, hence, non-Abelian. In particular, three non-Abelian planon fluxes decorate the original fracton flux; hence, this decorated fracton flux has a quantum dimension of $(\sqrt{2})^3$. This model is similar to that in Ref. [1], where the fracton is fundamentally immobile and also inextricably non-Abelian [7, 10]. We note that it is also possible to obtain non-Abelian fractons from gauging twisted Abelian theories as discussed in Ref. [10].

# 5 Summary and outlook

In this work, we construct exotic fracton models by taking decoupled Ising models with global subsystem symmetries and gauging a subgroup of the associated full symmetry group. We discovered that on gauging this composite symmetry, the flux sector is given by the composite of fluxes associated with the gauged versions of the decoupled models. In other words, the original fluxes in the gauged models bind to give the flux in the model with gauged composite symmetry. We also used the principle of remote detectability to demonstrate flux binding. In particular, we construct remote detection operators for detecting symmetry charges of the

composite symmetry and truncate them to find the bound flux.

The model obtained by gauging the composite symmetry can be equivalently obtained by condensing a composite of gauge charges in the gauged decoupled models. This composite gauge charge corresponds to the new minimal coupling obtained for the composite symmetry. We expect that such a condensation can be realized using a sequential linear-depth circuit. Since we consider only planar symmetries in this work, such a sequential linear-depth circuit would also be in-plane. In particular, starting from Model FP1 or FP2, we can obtain the gauged decoupled models by application of in-plane sequential linear-depth circuits that condense fractional excitations in foliations. The gauged models could, hence, be understood as generalized foliated fracton models where the notion of generalized foliation is described in Ref. [11]. Thus, in this sense of generalized foliation, our models are equivalent to the decoupled models. On the ungauged side, this corresponds to having a weak subsystem symmetry-protected topological phase (SSPT) [12] if we apply the notion of generalized foliation in the definition of SSPTs. Thus, we expect the models obtained by gauging strong SSPTs cannot be obtained using the construction presented in this work.

Following the picture of flux binding as composite charge condensation, we note that the constituent (gauged) X-cube model, used in the construction of Model FP1 (FP2), can also be obtained by a charge-condensation process in toric code layers, known as p-loop condensation [7,9]. Thus, the (gauged) ModelFP1 can be obtained by starting with stacks of toric code layers, performing p-loop condensation in a subset of them to obtain the X-cube model, and then performing the composite charge condensation mentioned earlier. Since the condensation in both the steps is of charges, we can perform them in any order and obtain the gauged Model FP1.

Gauging a composite symmetry can be considered a special case of gauging a subgroup of the symmetry group. Such gauging of subgroups of the symmetry group has been considered before. For instance, in the hybrid fracton models of Refs. [13] and [14], instead of gauging the $\mathbb{Z}_4$ global symmetry of layers of 2D $\mathbb{Z}_4$ toric codes, one can gauge a $\mathbb{Z}_2$ subgroup of the symmetry such that the pairs of $e^2$ across neighboring layers are condensed to obtain "hybrid toric code layers".

Throughout this work, we considered planar subsystem symmetries. An interesting future direction involves extending this paper's analysis to fractal symmetries and a combination of fractal and planar symmetries. A potential issue arises in obtaining geometrically local fracton models when gauging a combination of fractal and planar symmetries. In particular, it is not obvious how to choose composite subsystem symmetries such that the set of symmetric operators is generated by purely local terms. For all (subsystem) symmetries we consider in this paper, their symmetric operators can be locally generated. For example, in a 2D system with a global $\mathbb{Z}_2$ symmetry, the set of symmetric operators can be generated by an on-site transverse field term and a nearest-neighbor Ising coupling term. The gauge field DOF's are then placed at the location of the Ising terms. If, on the other hand, the symmetric operators are not locally generated, the gauging process would yield a non-local model. For example, consider a 2D system with line symmetries in both the $x$ and $y$ directions and try gauging the composite symmetry of the tensor product of symmetries on row $i$ and column $i$. In an $N \times N$ system, there are $N$ symmetry generators. All two-body Ising coupling terms between sites $(j, k)$ and $(k, j)$ are now symmetric. However, since this set of terms cannot be locally generated, the gauged model would be non-local. On the other hand, gauging composite symmetries can yield interesting $k$-local Hamiltonians corresponding to quantum low-density parity check (LDPC) codes [15]. In general, obtaining LDPC codes via gauging can be insightful in understanding the associated code properties from the perspective of the gauging duality. In order to obtain an LDPC code Hamiltonian, we would want, in the ungauged model, each qubit and each relation to support and involve constant number of $k$-local terms respectively.

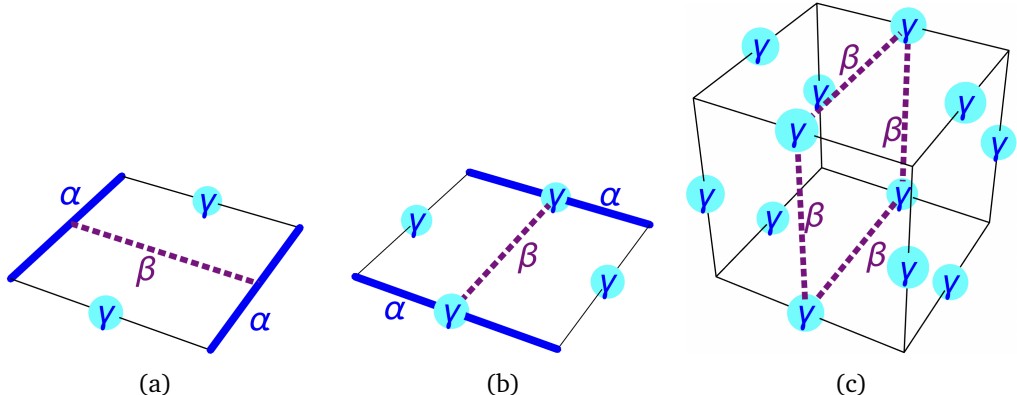

Figure 18: Model LP: the **E**-type flux terms. Drawn here are: (a) the $\mathbf{E}_p^{[z](x)}$, and (b) the $\mathbf{E}_p^{[z](y)}$ terms. The remaining $\mathbf{E}_p^{[x](\cdot)}$ and $\mathbf{E}_p^{[y](\cdot)}$ terms are similar. (c): one of the three relations between the original flux terms, which is the product of a cube flux term of the lineon-charge system and a plaquette flux term of the planon-charge system. It is obtained from multiplying the **E**-type flux terms around the faces of a cube. The other two relations can be obtained by rotations.

Whether this is possible or not would depend on the map from initial to final ungauged symmetric terms obtained after combining the symmetries.

## Acknowledgments

We are indebted to inspiring discussions with Xiuqi Ma.

**Funding information** A.V. is supported by the Ernest R. Roberts SURF fellowship. A.D., Z.W., and X.C. are supported by the National Science Foundation under award number DMR-1654340, the Simons collaboration on "Ultra-Quantum Matter" (grant number 651438), the Simons Investigator Award (award ID 828078) and the Institute for Quantum Information and Matter at Caltech. W.S. acknowledges support from the Simons collaboration on "Ultra-Quantum Matter" (grant number 651444). X.C. is also supported by the Walter Burke Institute for Theoretical Physics at Caltech.

## A  Model LP: Gauged Hamiltonian

In Sec. 4.2, we introduced a lineon-charge system that is the dual of the fracton-charge system used in Sec. 2.2. Gauging this model yields the X-cube model but now with the identification of the fracton superselection sectors as fluxes. We added three stacks of 2D planon charge systems to the model (shown in Fig. 15(a)) and gauged a composite symmetry with generators supported on both models to yield the model LP with decorated fracton superselection sectors. To illustrate the flux binding that leads to the decorated fracton superselection sectors, we constructed the remote detection operator (a membrane operator shown in Fig. 16(c)) of the fluxes in the model with gauged composite symmetry. In this appendix, we show the flux binding by writing the fully gauged Hamiltonian.

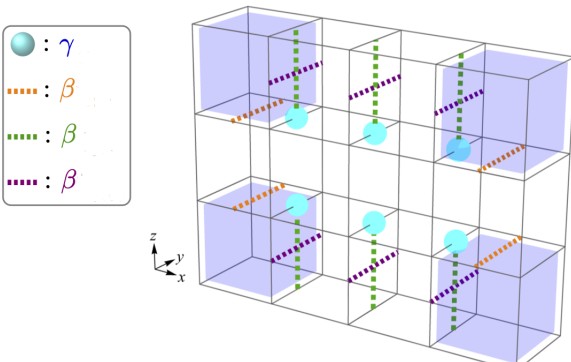

Figure 19: Model LP: the new fracton membrane operator where the fractons are created at the four corners, the blue cubes. The new fracton flux is a product of three original planon fluxes from three mutually perpendicular planon-charge systems together with the original fracton flux from the 3D lineon-charge system.

Following the procedure in Sec. 2, we add gauge fields to the coupling terms as shown in (b) and (c) of Fig. 15.

$$H^{LP} = -\sum_{x' \in P^x_{(i_x, i_x+1)}} \sum_{y' \in P^y_{(i_y, i_y+1)}} \sum_{z' \in P^z_{(i_z, i_z+1)}} \mathbf{\Gamma}^{(x',y',z')} - \sum_{\mu, \{i_\mu\}} \left( \sum_{p \in P^\mu_{(i_\mu, i_\mu+1)}} \mathbf{C}^\mu_p + \sum_{\nu, p \in P^\mu_{i_\mu}} \mathbf{E}^{[\mu](\nu)}_p \right)$$
$$- \text{(charge terms)}. \tag{A.1}$$

The $\mathbf{\Gamma}$-type flux terms are the original 12-body cube fluxes of the lineon-charge system. The $\mathbf{C}$-type flux terms are the original 4-body plaquette fluxes of the planon-charge system (shown in Fig. 6(a)). The $\mathbf{E}$-type flux terms, involving both the original minimal couplings and the new minimal couplings, are given in (a) and (b) of Fig. 18. Since these $\mathbf{E}$ terms do not contribute to the superselection sectors (since they can be converted to $\mathbf{C}$-type fluxes via a $\beta^x_{[\mu](\nu)}$ operator), we can set them to the identity. This yields relations between the original flux terms, e.g. (c) of Fig. 18. The product of 4 $\mathbf{E}$ terms around the faces of a cube equals the identity, but it is also equivalent to the product of a $\mathbf{\Gamma}$ and a $\mathbf{C}$ flux term. Using these constraints, we construct the logical operator for the new flux excitation in Fig. 19. The new flux is also a fracton since it is a composite of an original fracton together with three original planon fluxes, each from a different 2D system.

# B Connection to cage-net fracton models

In Sec. 4, we saw that Model FP1 (Sec. 2.2) can give rise to the cage-net fracton models [7,9]. The cage-net models are constructed using three stacks of 2D topological orders and then coupled with particle-loop (p-loop) condensation. Here, we show how the p-loop condensation can be understood as gauging a composite symmetry.

Consider two 'composite condensates' in the $z$-direction, i.e. the charges associated with the $\alpha_{(z)}$-coupling term, Fig. 5(a). And two 'composite condensates' in the $x$-direction. The product of these four terms is equal to four planon charges on the same face of a cube, Fig. 20. If we represent a planon charge by a line orthogonal to its plane, then we get a loop, i.e. the orange loop in Fig. 20. This loop is precisely the p-loop considered in Ref. [7,9]. Since we constructed this p-loop through the 'composite condensates', it is also condensed. Therefore, gauging the composite symmetry is the same as condensing the p-loop. We can then regard Model FP1 as the ungauged version of the cage-net fracton models.

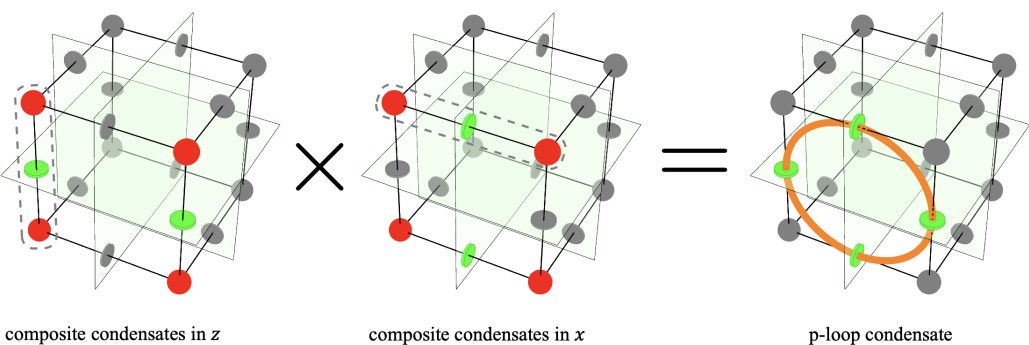

composite condensates in $z$ $\times$ composite condensates in $x$ $=$ p-loop condensate

Figure 20: Particle-loop (p-loop) condensation in Model FP1. The p-loop, the orange loop on the RHS, is generated by the 'composite condensates' on the LHS.

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
