# Peer review of "Composite subsystem symmetries and decoration of sub-dimensional excitations"

_SciPost Physics, doi:SciPost Phys. 17, 071 (2024)_

## Round 1 · Referee Report · Anonymous (Referee 1) · 2024-4-21

Report

This paper explores the gauge fluxes of gauge theories that arise from gauging different sub-symmetries of a global symmetry (subsystem 1 symmetry) x (subsystem 2 symmetry) x.... In particular, they show how the gauge fluxes that arise from gauging the entire symmetry group are related to those that arise by gauging diagonal subgroups. The paper is well organized and interesting to, at least, both the fracton/topological order community as well as the generalized symmetries community. I am therefore happy to recommend it for publication. However, I hope the authors will consider some of my comments and questions below, which could clarify parts of their paper and provide interesting questions for additions to this paper (or follow-up work).

Three questions I have regarding gauging: 1) How do the authors expect different types of gauging to affect their results (i.e., various types of twisted gauging implemented by conjugating with an SPT entangler before and after gauging)? 2) I believe the authors always assume the symmetry before gauging is not spontaneously broken (e.g., the starting model is a paramagnet). My understanding of why is that by gauging a non-SSB'd subsystem symmetry, you get a model with a dual subsystem symmetry that is SSB'd, which explains why you have subdimensional gauge charges. Do the authors know how things would change if their starting point differed, especially if the starting model was in a nontrivial SPT phase protected by the symmetry? 3 Do the authors know how the 't Hooft anomalies of the 0-form and subsystem symmetries affect their results? In particular, if the symmetry was anomalous and they gauged an anomaly-free subgroup, would they expect to get some sort of subsystem higher-groupoid or subsytem noninvertible symmetry?

For 2+1d abelian topological orders, the principle of remote detectability can be nicely stated in terms of 't Hooft anomalies of higher-form symmetries. Can the authors phrase their remote detectability principle using 't Hooft anomalies?

To improve clarify, could the authors distinguish between global and gauge symmetries, and global symmetry charges and gauge symmetry charges, more precisely in the introduction? I found the distinction clear in the main text and conclusion section, but I had difficulty following the terminology in the introduction. Relatedly, if the authors could clarify in the introduction what charges in which theory are being condensed (e.g., in the second paragraph), that would also make the introduction clearer.

Recommendation

Publish (easily meets expectations and criteria for this Journal; among top 50%)

  • validity: -
  • significance: -
  • originality: -
  • clarity: -
  • formatting: -
  • grammar: -

Author:  Avi Vadali  on 2024-07-29  [id 4664]

(in reply to Report 1 on 2024-04-21)

We want to thank the referee for the helpful report. In response to the comments in the report, we have:

  1. added the definition of RD (remote detection) on page 9 when it is first used

  2. replaced "Model B" with "Model FP2", consistent with other parts of the paper

  3. Regarding the extension to fractal symmetries, we have updated the last paragraph in the summary section as follows:

Throughout this work, we considered planar subsystem symmetries. An interesting future direction involves extending this paper's analysis to fractal symmetries and a combination of fractal and planar symmetries. A potential issue arises in obtaining geometrically local fracton models when gauging a combination of fractal and planar symmetries. In particular, it is not obvious how to choose composite subsystem symmetries such that the set of symmetric operators is generated by purely local terms. For all (subsystem) symmetries we consider in this paper, their symmetric operators can be locally generated. For example, in a 2D system with a global Z2 symmetry, the set of symmetric operators can be generated by an on-site transverse field term and a nearest-neighbor Ising coupling term. The gauge field DOF's are then placed at the location of the Ising terms. If, on the other hand, the symmetric operators are not locally generated, the gauging process would yield a non-local model. For example, consider a 2D system with line symmetries in both the x and y directions and try gauging the composite symmetry of the tensor product of symmetries on row i and column i. In an NxN system, there are N symmetry generators. All two-body Ising coupling terms between sites (j,k) and (k,j) are now symmetric. However, since this set of terms cannot be locally generated, the gauged model would be non-local. On the other hand, gauging such composite symmetries can yield interesting k-local Hamiltonians corresponding to quantum low-density parity check (LDPC) codes. In general, obtaining LDPC codes via gauging can be insightful in understanding the associated code properties from the perspective of the gauging duality. In order to obtain an LDPC code Hamiltonian, we would want, in the ungauged model, each qubit and each relation to support and involve constant number of k-local terms respectively. Whether this is possible or not would depend on the map from initial to final ungauged symmetric terms obtained after combining the symmetries.

---

## Round 1 · Referee Report · Anonymous (Referee 2) · 2024-4-30

Report

The paper investigates the gauging of composite subsystem symmetries, whose parent theories host charge excitations with distinct sub-dimensional mobilities, e.g., fractons and planons. Notably they find the new, composite fluxes depend on how the original symmetries are combined. The exposition is clear and pedagogical, walking the reader through their findings by contrasting two case examples (their Models FP1 and FP2). The topic is of great interest to researchers working in the topological order community, with special focus on fracton theories.

I am thus very pleased to recommend its publication in the present form. One point I think it could be minimally expanded is the extension to fractal symmetries commented in the conclusions. Maybe the authors could add one sentence or two pointing out possible challenges (if there are any) and how sensitive to details the new fluxes are. Finally, two minor textual details the authors may wish to consider are:

1- the acronym "RD" is already used in page 9, but only defined later (page 15 of main text);

2- in the opening sentence of Sec. III-B there is a reference to a model "B", which may cause some confusion. Consider clarifying whether this refers to a distinct model or if it is a typo.

Recommendation

Publish (easily meets expectations and criteria for this Journal; among top 50%)

  • validity: high
  • significance: high
  • originality: high
  • clarity: top
  • formatting: excellent
  • grammar: perfect

Author:  Avi Vadali  on 2024-07-29  [id 4663]

(in reply to Report 2 on 2024-04-30)

We want to thank the referee for the interesting comments. Here are our responses:

  1. The idea of `twisted gauging' is an interesting one which leads to interesting observations regarding 1+1D theories. However, in this paper, we mainly consider systems with planar symmetries, which upon gauging, turn into systems with long-range entanglement. Because of this, we cannot 'conjugate' the gauging process with an SPT entangler because the systems before and after the gauging have different (generalized) symmetries.

  2. Our conclusion in this paper applies to trivial and symmetry-protected paramagnets alike. In particular, the mobility of the gauge charges does not depend on whether the starting point is a trivial or non-trivial SPT.

  3. If we gauge a non-anomalous subgroup of an anomalous symmetry, a simple case is that the gauged theory breaks the remaining part of the anomalous symmetry. It is not obvious to us how groupoid or noninvertible symmetries can arise. That is definitely an interesting question.

  4. The interpretation of remote detectability in terms of 't Hooft anomalies is an interesting point. The principle should be similarly applicable to subsystem symmetries, although we will need to make use of a more sophisticated version of field theory as discussed in for example arXiv:2004.00015, arXiv:2008.03852, arXiv:2112.05726.

  5. Thanks for pointing out the ambiguity in our terminology. We have modified the introduction section to make clear the distinction between symmetry charges and gauge charges. In particular, in the second paragraph of the Introduction, we discuss the coupling of 2 planar Z2 gauge theories. Here we clarify that forming a composite symmetry between the two Z2 symmetry generators makes the composite of the two symmetry charges no longer a symmetry charge. Additionally, we clarify that the gauge charge pair is condensed after gauging the composite Z2 symmetry.

---

## Round 2 · Author Response

To referee 1
We want to thank the referee for the helpful report. In response to the comments in the report, we have:
-
added the definition of RD (remote detection) on page 9 when it is first used
-
replaced "Model B" with "Model FP2", consistent with other parts of the paper
-
Regarding the extension to fractal symmetries, we have updated the last paragraph in the summary section as follows:
Throughout this work, we considered planar subsystem symmetries. An interesting future direction involves extending this paper's analysis to fractal symmetries and a combination of fractal and planar symmetries. A potential issue arises in obtaining geometrically local fracton models when gauging a combination of fractal and planar symmetries. In particular, it is not obvious how to choose composite subsystem symmetries such that the set of symmetric operators is generated by purely local terms. For all (subsystem) symmetries we consider in this paper, their symmetric operators can be locally generated. For example, in a 2D system with a global Z2 symmetry, the set of symmetric operators can be generated by an on-site transverse field term and a nearest-neighbor Ising coupling term. The gauge field DOF's are then placed at the location of the Ising terms. If, on the other hand, the symmetric operators are not locally generated, the gauging process would yield a non-local model. For example, consider a 2D system with line symmetries in both the x and y directions and try gauging the composite symmetry of the tensor product of symmetries on row i and column i. In an NxN system, there are N symmetry generators. All two-body Ising coupling terms between sites (j,k) and (k,j) are now symmetric. However, since this set of terms cannot be locally generated, the gauged model would be non-local. On the other hand, gauging such composite symmetries can yield interesting k-local Hamiltonians corresponding to quantum low-density parity check (LDPC) codes. In general, obtaining LDPC codes via gauging can be insightful in understanding the associated code properties from the perspective of the gauging duality. In order to obtain an LDPC code Hamiltonian, we would want, in the ungauged model, each qubit and each relation to support and involve constant number of k-local terms respectively. Whether this is possible or not would depend on the map from initial to final ungauged symmetric terms obtained after combining the symmetries.
To referee 2
We want to thank the referee for the interesting comments. Here are our responses:
-
The idea of `twisted gauging' is an interesting one which leads to interesting observations regarding 1+1D theories. However, in this paper, we mainly consider systems with planar symmetries, which upon gauging, turn into systems with long-range entanglement. Because of this, we cannot 'conjugate' the gauging process with an SPT entangler because the systems before and after the gauging have different (generalized) symmetries.
-
Our conclusion in this paper applies to trivial and symmetry-protected paramagnets alike. In particular, the mobility of the gauge charges does not depend on whether the starting point is a trivial or non-trivial SPT.
-
If we gauge a non-anomalous subgroup of an anomalous symmetry, a simple case is that the gauged theory breaks the remaining part of the anomalous symmetry. It is not obvious to us how groupoid or noninvertible symmetries can arise. That is definitely an interesting question.
-
The interpretation of remote detectability in terms of 't Hooft anomalies is an interesting point. The principle should be similarly applicable to subsystem symmetries, although we will need to make use of a more sophisticated version of field theory as discussed in for example arXiv:2004.00015, arXiv:2008.03852, arXiv:2112.05726.
-
Thanks for pointing out the ambiguity in our terminology. We have modified the introduction section to make clear the distinction between symmetry charges and gauge charges. In particular, in the second paragraph of the Introduction, we discuss the coupling of 2 planar Z2 gauge theories. Here we clarify that forming a composite symmetry between the two Z2 symmetry generators makes the composite of the two symmetry charges no longer a symmetry charge. Additionally, we clarify that the gauge charge pair is condensed after gauging the composite Z2 symmetry.

---

## Round 2 · List of Changes

The list of changes is as follows:
1. added the definition of RD (remote detection) on page 9 when it is first used
2. replaced "Model B" with "Model FP2", consistent with other parts of the paper
3. Regarding the extension to fractal symmetries, we have updated the last paragraph in the summary section as follows:
Throughout this work, we considered planar subsystem symmetries. An interesting future direction involves extending this paper's analysis to fractal symmetries and a combination of fractal and planar symmetries. A potential issue arises in obtaining geometrically local fracton models when gauging a combination of fractal and planar symmetries. In particular, it is not obvious how to choose composite subsystem symmetries such that the set of symmetric operators is generated by purely local terms. For all (subsystem) symmetries we consider in this paper, their symmetric operators can be locally generated. For example, in a 2D system with a global Z2 symmetry, the set of symmetric operators can be generated by an on-site transverse field term and a nearest-neighbor Ising coupling term. The gauge field DOF's are then placed at the location of the Ising terms. If, on the other hand, the symmetric operators are not locally generated, the gauging process would yield a non-local model. For example, consider a 2D system with line symmetries in both the x and y directions and try gauging the composite symmetry of the tensor product of symmetries on row i and column i. In an NxN system, there are N symmetry generators. All two-body Ising coupling terms between sites (j,k) and (k,j) are now symmetric. However, since this set of terms cannot be locally generated, the gauged model would be non-local. On the other hand, gauging such composite symmetries can yield interesting k-local Hamiltonians corresponding to quantum low-density parity check (LDPC) codes. In general, obtaining LDPC codes via gauging can be insightful in understanding the associated code properties from the perspective of the gauging duality. In order to obtain an LDPC code Hamiltonian, we would want, in the ungauged model, each qubit and each relation to support and involve constant number of k-local terms respectively. Whether this is possible or not would depend on the map from initial to final ungauged symmetric terms obtained after combining the symmetries.
4. We have modified the introduction section to make clear the distinction between symmetry charges and gauge charges. In particular, in the second paragraph of the Introduction, we discuss the coupling of 2 planar Z2 gauge theories. Here we clarify that forming a composite symmetry between the two Z2 symmetry generators makes the composite of the two symmetry charges no longer a symmetry charge. Additionally, we clarify that the gauge charge pair is condensed after gauging the composite Z2 symmetry.

---

## Editorial Decision

published